# Specific Interactions between Human Norovirus and Environmental Matrices: Effects on the Virus Ecology

**DOI:** 10.3390/v11030224

**Published:** 2019-03-05

**Authors:** Mohan Amarasiri, Daisuke Sano

**Affiliations:** 1Department of Civil and Environmental Engineering, Graduate School of Engineering, Tohoku University, Aoba 6-6-06, Aramaki, Aoba-ku, Sendai, Miyagi 980-8579, Japan; daisuke.sano.e1@tohoku.ac.jp; 2Department of Frontier Science for Advanced Environment, Graduate School of Environmental Studies, Tohoku University, Aoba 6-6-06, Aramaki, Aoba-ku, Sendai, Miyagi 980-8579, Japan

**Keywords:** human norovirus, histo-blood group antigens, specific interactions, wastewater, *Enterobacter*

## Abstract

Human norovirus is the major cause of non-bacterial epidemic gastroenteritis. Human norovirus binds to environmental solids via specific and non-specific interactions, and several specific receptors for human norovirus have been reported. Among them, histo-blood group antigens (HBGA) are the most studied specific receptor. Studies have identified the presence of HBGA-like substances in the extracellular polymeric substances (EPS) and lipopolysaccharides (LPS) of human enteric bacteria present in aquatic environments, gastrointestinal cells, gills, and palps of shellfish, and cell walls, leaves, and veins of lettuce. These HBGA-like substances also interact with human norovirus in a genotype-dependent manner. Specific interactions between human norovirus and environmental matrices can affect norovirus removal, infectivity, inactivation, persistence, and circulation. This review summarizes the current knowledge and future directions related to the specific interactions between human norovirus and HBGA-like substances in environmental matrices and their possible effects on the fate and circulation of human norovirus.

## 1. Introduction

Human norovirus is the major cause of non-bacterial epidemic gastroenteritis. More than 18% of worldwide gastroenteritis cases are attributed to it [1,2]. According to recent estimates, the direct health costs of norovirus gastroenteritis are around $4.2 billion, while the societal costs are around $60 billion [3]. Noroviruses are single-stranded RNA viruses with a genome of ~7.7 kb in size, consisting of three open reading frames (ORFs) [4]. There are seven reported genogroups for human noroviruses (GI–GVII), and among them, GI, GII, and GIV can cause infections in humans [5]. Norovirus strains belonging to these genogroups are further subdivided into capsid and polymerase genotypes [5,6].

Up to 1.6 × 10^12^ genome copies of human norovirus per gram of feces are discharged from infected individuals [7]. Sewage, containing these norovirus particles, is discharged to natural water bodies or end up in wastewater treatment plants. Some wastewater treatment plants are not capable of completely removing human norovirus [8,9]. Therefore, wastewater treatment plant effluents, discharged to natural water bodies, can also become a source of human norovirus [10]. A recent norovirus outbreak caused by consuming bottled water was attributed to spring aquifer contamination by sewage [11]. Agricultural irrigation using improperly treated wastewater can expose humans to norovirus and transfer norovirus to produce, which can ultimately lead to outbreaks [12,13,14,15,16,17]. Contamination of water in shellfish cultivation areas by sewage can lead to accumulation of norovirus in the shellfish digestive tissues and consequently to outbreak situations [18]. Recreational water-related norovirus outbreaks due to lake water contamination by beach users were reported [19,20].

Viruses in the environment are frequently associated with particulate matter or other surfaces [21]. Norovirus GI and GII were shown to attach to large settleable particles (>180 µm), smaller suspended particles (>0.45 µm), and colloidal particles in a waste stabilization pond [22]. There were differences in the adsorption and aggregation of GI.I and GII.4 virus-like particles (VLPs) depending on the solution chemistry [23,24]. During the evaluation of virus removal performance by a pilot-scale membrane bioreactor (MBR), the association of norovirus with mixed liquor suspended solids (MLSS) was comparatively higher than that of enteroviruses [25]. Another full-scale MBR study reported that 91% of norovirus GII particles inside the reactor were attached to mixed liquor solids [26]. The difference between the concentration of adsorbed viruses in to MLSS and the concentration present in the liquid phase was the highest for norovirus GI, followed by sapovirus, and norovirus GII, while rotavirus did not display any significant differences [24].

The interactions between enteric viruses and environmental matrices can be separated in to non-specific interactions and specific interactions [27]. Non-specific interactions mainly consist of electrostatic interactions and hydrophobic interactions, whereas specific interactions require receptor–ligand assemblies [21,27,28]. Electrostatic interactions between environmental solids and enteric viruses can be explained using the Derjaguin, Landau, Verwey, and Overbeek (DLVO) theory [21]. A study on the effects of virus surface characteristics on virus removal reported that, depending on the number of hydrophobic amino acid groups in the external capsid of a bacteriophage, it can interact more freely with the hydrophobic portions of bacterial flocs and the extracellular polymeric substance (EPS) in MLSS [29].

However, the possible effects of specific interactions between human norovirus and environmental matrices on norovirus ecology have not been completely evaluated yet. This review summarizes the current knowledge on specific interactions between human norovirus and specific environmental receptors and their effects on norovirus ecology. Important research questions, which need to be answered to further understand the environmental norovirus ecology, are highlighted.

## 2. Specific Adsorbents for Human Noroviruses

Among the number of specific adsorbents for human norovirus reported until now, histo-blood group antigens (HBGAs) are the most studied adsorbent. HBGAs are complex carbohydrates present in the outer part of N- or O- linked glycans of glycoproteins and glycolipids [30]. ABH and Lewis HBGA families are present, and the secretor or non-secretor status is determined by the presence of fucosyltransferase2 (FUT2) [31]. The biosynthesis of HBGAs occurs by sequential addition of monosaccharides to the type-1 or type-2 disaccharide precursors mediated by FUT2 (α-1,2 fucosyltransferase), FUT3 (α-1,3 or α-1,4 fucosyltransferase), and A and B transferases [31,32,33]. HBGAs can be found on the surfaces of red blood cell and mucosal epithelial cells or are available as free oligosaccharides in bodily fluids like milk, saliva, and blood of secretor-positive (Se+) individuals [31,34]. Different norovirus strains have different HBGA-recognition profiles; up to now, eight distinct binding patterns have been recognized [35,36,37]. The P2 subdomain of the norovirus capsid is directly responsible for receptor recognition [38]. A study was conducted to determine the precise locations and binding modes of HBGAs on the viral capsids using a recombinant P protein of GII.4 VA387 strain cocrystallized with synthetic A or B trisaccharides [38]. The results revealed that both A and B saccharides strongly interact with P protein of VA387 [39]. α-fucose plays a central role in norovirus–receptor interactions, while β-galactose may not be crucial [39]. The cavity which binds with α-fucose is formed by the β5 strand and residues S441, G442, and Y443 in one monomer and S343, T344, R345, and D374 in the other monomer [37,39]. The interface between P protein and fucose is dominated by hydrogen bonds. A recent study reported that not only the presence of a binding epitope but also the orientation of the receptor is critical for norovirus binding [40]. Norovirus GII.4 strains that have evolved towards pandemic strains show increased relative affinity for HBGAs, and some of the recent variants have a broader host spectrum, including Lewis-positive non-secretors [41,42].

In addition to HBGAs, human norovirus has shown to attach to heparan sulphate [43], sialylated glycans [44], virus binding proteins recovered from activated sludge [45,46], and norovirus attachment proteins in mammalian cells [47]. A study by Gandhi et al. [48] reported specific binding of GI to the surface of romaine lettuce, cilantro, and iceberg lettuce via non-HBGA receptors which are not characterized yet. Almand et al. [49] observed the binding of noroviruses GI.6, GII.4 New Orleans, GII.4 Sydney, and Tulane virus to a selected group of bacteria representing the human gut microbiome and found that all bacterial strains interacted with the virus strains. HBGAs were suspected to be the specific receptor, even though the specific receptors responsible for norovirus binding are yet to be identified. Several substances, such as human milk oligosaccharides [50], glycerol [51], and tannic acids [52], are reported to inhibit the interaction between HBGA-like substances and human norovirus. Extensive reviews on the receptors interacting with human norovirus are available elsewhere [53,54].

Human norovirus surrogates are also shown to establish specific interactions with various receptors. Murine norovirus (MNV) can bind to GD1a located in the terminal sialic acids on murine macrophages and was confirmed to have a region topologically similar to the HBGA-binding site of norovirus VA387 strain [55,56]. Studies have confirmed the necessity of CD300lf for the binding and replication of MNV in cell lines [57,58]. Tulane virus binds to A-type 3 and B-type HBGAs and also recognizes sialic acids for cell attachment [59,60].

## 3. Specific Interactions between HBGA-Like Substances and Human Norovirus

HBGA-like substances can be found in many environmental materials like bacteria, leafy greens, and shellfish (Table 1) [61,62,63,64]. In 2013, Miura et al. [61] reported for the first time on the isolation of an HBGA-positive bacterial strain of *Enterobacter cloacae* SENG-6 from the fecal sample of a healthy adult using anti-blood group antibodies. Studies have reported the binding of human norovirus to HBGA-like substances located in the gastrointestinal epithelial cells of oysters, mussels, and clams [62,64,65,66,67]. HBGA-like substances present in the cell wall of lettuce have shown to specifically interact with human norovirus [63].

HBGA-like substances present in the EPS of *E. cloacae* SENG-6 bound to GI.1 (8fIIa) and GII.6 VLPs, while GI.1 (W375a) VLPs which do not have HBGA binding ability cannot bind successfully. The specificity of this interaction was confirmed using enzymatic cleavage of terminal N-acetyl-galactosamine, which led to reduced binding of GI.1 (8fIIa) [61]. GI.1 (Norwalk virus) and GII.4 (Dijon) VLP binding ability to *Escherichia coli* LMG8223 and *E. coli* LFMFP861 correlated with the HBGA expression profile of those bacterial strains [68]. H-type HBGA-like substances present in the tissues of Romaine lettuce leaves were identified as the receptors responsible for GII.4 norovirus binding. Enzymatic digestion of the cell wall increased GII.4 binding, while pretreatment of the cell wall materials with α-1,2-fucosidase diminished the interaction [63]. B-type HBGA-like substances containing *E. coli* O86:H2 bound to Tulane virus, whereas HBGA-negative *E. coli* K-12 did not bind efficiently. On the basis of bacterial-capture-RT-qPCR results, the binding capacity of *E. coli* O86:H2 was five times higher than that of *E. coli* K-12. Pretreatment with free HBGA reduced the binding capacity of Tulane virus to *E. coli* O86:H2, emphasizing the specific nature of this interaction [69]. Wang et al. [70] engineered a novel system which can present human norovirus VP1 on the bacterial surface and confirmed the interactions between human norovirus and HBGA-like substances present on the leaves and veins of romaine lettuce. The specificity of the interaction was elucidated by a competitive adsorption experiment using HBGAs from human saliva.

## 4. Effects of the Interactions with HBGA-Like Substances on Norovirus Ecology

In this section, the effects of specific interactions between human norovirus and HBGA or HBGA-like substances and how these interactions affect norovirus removal, infectivity, persistence, survival, and circulation are discussed on the basis of the results from recent studies.

### 4.1. Norovirus Removal

Amarasiri et al. [71] evaluated the removal of human norovirus GII.3, GII.4, and GII.6 VLPs by microfiltration in the presence of HBGA-positive *E. cloacae* SENG-6 (HBGA-like substances located in EPS) and *E. coli* O86:K61:B7 (HBGA-like substances located in lipopolysaccharides (LPS)) and observed genotype-dependent differences in the norovirus removal efficiency. *E. cloacae* SENG-6 significantly contributed to the removal of VLPs compared to *E. coli* O86:K61:B7. However, the removal of EPS from both bacterial strains produced opposite results, displaying the importance of the localization of HBGA-like substances in virus removal (Figure 1). A subsequent study used rotavirus HAL1166 strain which interacts with A-type HBGA and can be cultivated in large volumes as a surrogate for human norovirus in a lab-scale cross-flow membrane filtration study to evaluate the contribution of specific interactions in norovirus removal [73,74]. Trypsin treatment of rotavirus was used to differentiate between specific and non-specific interactions with HBGA-like substances of *E. cloacae* SENG-6. During the filtration test, the amount of trypsin-treated HAL1166 in the filtrate displayed a decreasing trend, while that of the untreated rotavirus HAL1166 varied abruptly. (Figure 2) [73,74,75].

### 4.2. Norovirus Cell Attachment and Infectivity

The contributions of commensal bacteria on cell attachment and infectivity of norovirus have been studied using the norovirus GII.4-Sydney strain, GII.4/2006b strain, and surrogates like MNV, norovirus P-particles, and Tulane virus.

The filtration of a stool sample containing norovirus GII.4-Sydney strain resulted in reduced attachment of GII.4 to BJAB B cells as measured by the infection. However, the incubation of the filtered stool sample with H-type HBGA expressing *E. cloacae* (ATCC 13047) displayed a dose-dependent infectivity restoration which was significantly higher (*p* < 0.001) than in the sample without *E. cloacae* (ATCC 13047). A similar increase of infectivity was observed when using synthetic H-antigen, confirming that the specific interactions between human norovirus and HBGA-like substances present in enteric bacteria can facilitate the infection of B cells [76].

In another study, the intestinal microbiota of C57BL/6J mice were depleted using an antibiotic cocktail. After that, the MNV strain CR6 was orally administered to the microbiota-depleted mice, and fecal virus shedding and the levels of virus in intestinal tissues were determined. After 3 and 14 days from infection, significantly low amounts (*p* < 0.001) of MNV were observed in fecal pellets, ileum, colon, and mesenteric lymph nodes (MLN), confirming that persistent MNV infection was prevented by the antibiotics. Flushing out the antibiotics from the mice did not restore CR6 infection. However, the transplantation of feces collected from mice not treated with antibiotics restored MNV CR6 infection, while fecal transplants from antibiotic-treated mice did not, confirming the role of the intestinal microbiota in persistent norovirus infection [77]. IFN-λ has been shown to play a major role in controlling persistent MNV infection because mice lacking IFN-λ induction or signaling pathway components are infected even at low MNV doses, regardless of the presence of microbiota. Therefore, it is suggested that commensal microbiota interaction with MNV is suppressing IFN-λ production, leading to persistent MNV infection [77,78,79,80].

Gnotobiotic pigs colonized with *E. cloacae* (ATCC 13047) and inoculated with the human norovirus GII.4/2006b strain showed significantly lower cumulative virus shedding (*p* < 0.05) 7 and 10 days after infection compared to uncolonized pigs. Similarly, significantly lower (*p* < 0.05) virus titers were observed in the duodenum and ileum of *E. cloacae*- (ATCC 13047) colonized pigs compared to controls at 3 days post-infection, showing that *E. cloacae* (ATCC 13047) colonization had an inhibitory effect on human norovirus shedding. The immunohistochemistry analysis of viral antigens in B cells from ileum found no signal co-localization, suggesting B cells were not the human norovirus target in gnotobiotic pigs. Furthermore, significantly larger and more developed ileal Payer’s patches were observed in the pigs inoculated with *E. cloacae* (ATCC 13047), providing evidence that gut immunity was enhanced by the enteric bacteria [81].

The presence of A-, B-, and H-type HBGA-like substances in lactic acid bacterial strains (LAB) isolated from human feces has been confirmed, and it is suggested that the interactions between LAB and HBGA expressed in the intestinal mucosa may aid in colonizing the gut [82,83]. By using norovirus P-particles having the same antigenic and HBGA-binding profiles as their parental NoVLPs, Rubio-del-Campo et al. [84] have shown that both GI.1 and GII.4 P-particles bind to bacterial strains, including probiotics, to a different extent. *E. coli* Nissle1917, which is Gram-negative, displayed the lowest binding capacity. Gram-positive bacterial strains with higher contents of peptidoglycan- and teichoic acids-containing carbohydrates were suggested to have a higher P-particle attachment capacity. Therefore, probiotic bacteria can counteract the infection by interacting with norovirus particles or by interacting with HBGAs.

Interactions of Tulane virus with both HBGA-positive *E. coli* O86:H2 or HBGA-negative *E. coli* K-12 contributed to a reduced binding between the virus and LLC-MK2 cells and, consequently, limited virus replication [69].

### 4.3. Norovirus Persistence, Survival, and Circulation in the Environment

Evidence regarding the presence of human norovirus in wastewater treatment plant effluents, surface water, and reclaimed water used for agricultural irrigation is increasing [10,12,85]. Many studies have been conducted to evaluate norovirus persistence in water environments and how abiotic environmental factors like pH, temperature, and light influence virus inactivation [86,87,88,89].

However, studies have shown that HBGA–norovirus interactions help noroviruses to escape abiotic stresses and survive. In the presence of HBGA-negative bacteria and in the absence of any bacteria, the immunoreactivities of norovirus GI.1 and GII.4 VLPs were decreased after 2 min of heating at 90 °C. However, in the presence of *E. coli* LMG8223 and *E. coli* LFMFP861, the integrity of VLPs was not altered by heat treatment, and the receptor binding ability was increased [68]. Changes in relative humidity (RH) have shown to affect the binding of GII.4 Osaka variant (Cairo4 strain) NoVLPs to saliva from A, B, H, and non-secretors. At extreme RH levels (10% and >85%), binding was preserved. [90]. Greenhouses used in cultivating fresh produce like lettuce are maintained at high RH values which promote HBGA-norovirus binding and persistence [91]. Shellfish can concentrate and bioaccumulate human norovirus in their tissues via HBGA-like substances in a strain-dependent manner, which explains the GI bias in shellfish-related norovirus outbreaks [64,92,93]. Human norovirus can survive inside shellfish for over four weeks, and even boiling for three minutes cannot completely eliminate norovirus particles from mussels [18,94,95,96].

The above studies provide evidence about how noroviruses can survive and persist in water environments and confirm the role of leafy greens and shellfish as norovirus transmission vehicles. Since improperly treated wastewater effluents and irrigation water act as a source of human norovirus, the development of improved sewage treatment facilities that reduce virus discharge into natural waters can contribute to dicrease norovirus transmission and circulation in water environments [97,98].

However, the attachment to HBGA-positive *E. coli* O86:H2 or HBGA-negative *E. coli* K-12 strain did not protect Tulane virus from heat denaturation in partial (56 °C, 10 min) or complete (56 °C, 30 min) inactivation conditions as observed for VLPs [69], even though Tulane virus binds to HBGA [68,69]. The main mechanisms of virus inactivation by heat are disruption of virus capsid integration and viral protein denaturation [99]. Even though NoVLPs have been shown to be more heat-stable than Tulane virus, both NoVLP and Tulane virus maintained their receptor binding ability after heat treatment (below 100 °C). However, heat treatment lead to loss of infectivity for Tulane virus [99]. Therefore, different detection methods used in different studies may result in different outcomes. These results emphasize the importance of further studies to understand the role of specific interactions between environmental matrices and human norovirus in the virus thermal stability.

Available research results evince the presence of HBGA-like substances in many environmental matrices. Moreover, the norovirus-binding carbohydrates present in environmental materials such as lettuce were confirmed to be HBGA by subsequent studies [63,100]. Therefore, the prevalence of HBGA in various environmental matrices should be further elucidated. Moreover, a better understanding of the contribution of HBG—norovirus specific interactions in aquatic environments to norovirus ecology may provide valuable insights to identify measures to reduce the disease burden caused by human norovirus.

## 5. Further Studies

Specific interactions between human norovirus and environmental materials are genotype-specific [71]. Studies have confirmed the presence of multiple norovirus genotypes in river water, wastewater effluents, and estuarine sediments in many instances [101,102,103,104,105]. Therefore, genotype-specific detection, quantification, and genotyping tools for norovirus will provide an excellent platform to evaluate and monitor properties such as removal, survival, persistence, inactivation, and disinfection resistance of different norovirus genotypes in water environments. Currently, RT-qPCR primers have been developed for the detection and quantification of human norovirus genotypes GII.3, GII.4, GII.6, and GII.17 [106]. However, the dominancy of a particular norovirus genotype can differ depending on the season, and therefore, it will be valuable to develop genotype-specific primers for other norovirus genotypes [107,108].

Many studies have shown that the virus–solid particle interactions can interrupt the disinfection processes [109,110]. Free and particle-associated noroviruses have provided different results in disinfection studies [111,112]. Since disinfection is a major step in the water treatment/reclamation process and non-specific and specific interactions have different characteristics, a study on the disinfection of human noroviruses specifically adsorbed onto environmental matrices will provide valuable insights for improving the disinfection process.

A number of specific receptors for norovirus and substances which can inhibit norovirus–HBGA interaction have already been identified [53]. It is reasonable to expect that new specific environmental receptors and interaction inhibitors for human noroviruses will be found. The identification of new environmental receptors and inhibitors will shed light on the behavior of human norovirus in water environments.

## Figures and Tables

**Figure 1 viruses-11-00224-f001:**
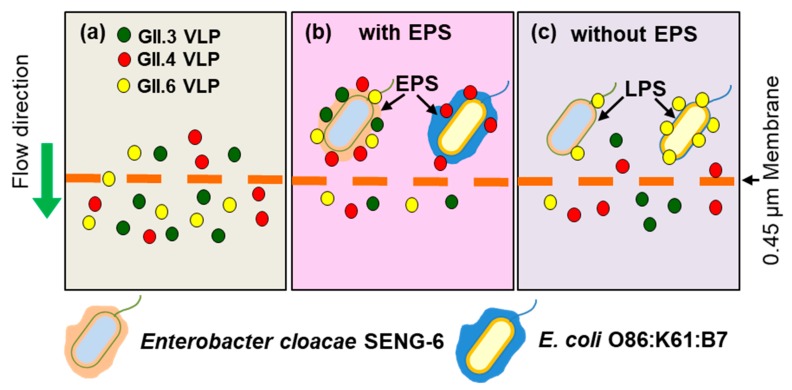
Removal of human norovirus GII3, GII.4, and GII.6 virus-like particles (VLPs) in the presence of histo-blood group antigen (HBGA)-positive *E. cloacae* SENG-6. (**a**) In the absence of SENG-6, all VLPs (~20–30 nm) pass through the microfiltration membrane (0.45 µm) without any obstruction; (**b**) In the presence of HBGA-positive SENG-6, all types of VLPs were retained, while HBGA-positive *E. coli* O86:B7 was able to bind only to GII.4 VLPs; (**c**) After the removal of EPS, SENG-6 lost the GII.6 VLP binding ability, while the VLP binding ability of *E. coli* O86:B7 increased.

**Figure 2 viruses-11-00224-f002:**
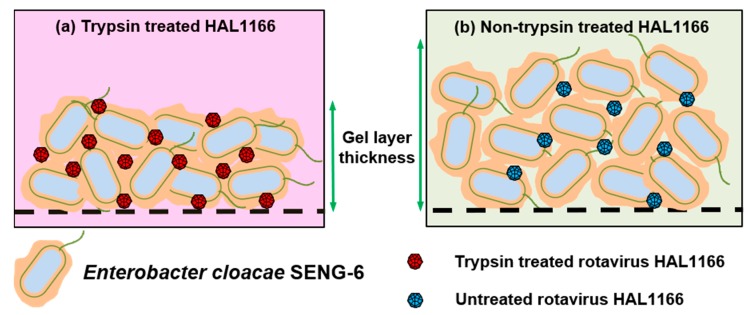
Development of a gel/cake layer on a microfiltration membrane surface during the filtration of a mixture of *E. cloacae* SENG-6 and rotavirus HAL1166 used as a surrogate for human norovirus. (**a**) The gel/cake layer developed by SENG-6+trypsin-treated rotavirus HAL1166 was less thick; (**b**) The gel/cake layer developed by SENG-6+untreated rotavirus HAL1166 had thicker and larger porosity.

**Table 1 viruses-11-00224-t001:** Specific interactions between human norovirus and histo-blood group antigens (HBGA)-like substances present in environmental matrices. EPS, extracellular polymeric substances; LPS, lipopolysaccharides.

Cell/Tissue	HBGA Activity	Location of HBGA-Like Substances	Interacting Norovirus Strains	Ref
*Enterobacter cloacae* SENG-6	A, B, H	EPS	GI.7, GII.3, GII.6 GII.4 (DenHaag 2006b)	[61]
*Escherichia coli* O86:K61:B7	B	LPS	GII.6	[71]
*E. coli* LMG8223	A, B, H, Le^a^, Le^b^, Le^x^, Le^y^		GI.1, GII.4 (Dijon 1996)	[68]
*E. coli* LFMFP861	B, Le^a^			
*E. coli* LFMFP289	B, Le^b^			
*Enterobacter aerogenes*	Le^a^, Le^b^, Le^y^		GI.1	
*Clostridium difficile*	Le^a^		GI.1	
*E. coli* O86:H2	B		Tulane virus	[69]
Romaine lettuce	A, B, H	Leaf	GII.4 (Sydney 2012)	[70]
	A, B, H	Vein	GII.4 (Sydney 2012)	
Lettuce	H	Cell wall	GII.4 (DenHaag 2006b)	[63]
*Crassostrea virginica* oysters	A	Gastrointestinal cells	GI.1 (8FIIa)	[62]
*Crassostrea virginica* oysters	A, H	Gastrointestinal cells	GI.1 (8FIIa) US 95/96 (VA387) GII.9 (VA207)	[65]
*Crassostrea sikamea* oysters	A, H			
*Crassostrea gigas* oysters	A, H			
*Venerupis japonica* clams	A	Gastrointestinal cells	GI.1 (8FIIa) GII.9 (VA207)	
*Mytilis edulis* mussels	A	Gastrointestinal cells	GI.1 (8FIIa) GII.9 (VA207)	
*Crassostrea gigas* oysters	A	Digestive tissues and palps	GI.1 (West Chester) GII.4 (Sydney 2012)	[66]
	H	Digestive tissues, gills and palps		
*Crassostrea gigas* oysters	A (100%)	Gut	GII.4 (DenHaag 2006b) GII.4 (US 95/96)	[72]
	A (61%) Le^b^ (91%)	Gills		

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
