# Peer review of "Specific Interactions between Human Norovirus and Environmental Matrices: Effects on the Virus Ecology"

_viruses, 2019, doi:10.3390/v11030224_

Round 1
Reviewer 1 Report
Amarasiri, M. and Daisuke Sano. Specific interactions between human norovirus and environmental matrices: effects of virus ecology. Viruses. 2019
In the manuscript, Amarasiri and Sano provide a detailed review of the current research regarding norovirus interactions with histo-blood group antigens (HBGAs) present in the environment. Although the review effectively summarizes the current knowledge in the field, a few things need to be addressed before publication.
Major Comments:
1. Overall, this reviewer found it hard to understand the importance of each section in the manuscript. I think readers would benefit if the authors added additional text providing better context for each section. For example, a paragraph at the beginning of the text detailing where in the environment norovirus is typically found and how this relates to human infections would be helpful. This would help readers better understand the context of norovirus interactions with leafy greens, removal from wastewater, etc.
2. Line 64: “Specific absorbents for human noroviruses.” The authors use the word absorbent throughout section 2 of the text. Absorbent usually defined as something that soaks up a liquid. In my opinion, this may not be the best word to describe binding of these viruses to specific receptors such as HGBAs.
3. In section 2, in the paragraph starting at line 82, since the authors describe some of the host cellular receptors for norovirus, they should also include the recent discovery of the CD300lf as a receptor for murine norovirus (Orchard RC et al. Science 2016).
4. Figure 1: Is the biosynthesis of HBGAs relevant to binding to norovirus and worthy of a figure in this review? I think a better figure would be to describe what specific environmental materials that contain HBGAs and their ability to bind to norovirus.
5. Figure 3: Having a figure dedicated to rotavirus seems outside of the scope of this review.
Minor:
1. There are many minor grammar errors throughout the text that need to be addressed.
Author Response
Major Comments:
Comment: Overall, this reviewer found it hard to understand the importance of each section in the manuscript. I think readers would benefit if the authors added additional text providing better context for each section. For example, a paragraph at the beginning of the text detailing where in the environment norovirus is typically found and how this relates to human infections would be helpful. This would help readers better understand the context of norovirus interactions with leafy greens, removal from wastewater, etc.
Response: Thank you very much for the comment. We have added a paragraph explaining the occurrence and transmission of human norovirus in water environments (line 36-47).
Comment: Line 64: “Specific absorbents for human noroviruses.” The authors use the word absorbent throughout section 2 of the text. Absorbent usually defined as something that soaks up a liquid. In my opinion, this may not be the best word to describe binding of these viruses to specific receptors such as HGBAs.
Response: We used the word “adsorbent” in our manuscript with the meaning “a substance that adsorbs another”.
Comment: In section 2, in the paragraph starting at line 82, since the authors describe some of the host cellular receptors for norovirus, they should also include the recent discovery of the CD300lf as a receptor for murine norovirus (Orchard RC et al. Science 2016).
Response: We added that information and modified the manuscript (line 109-114).
Comment: Figure 1: Is the biosynthesis of HBGAs relevant to binding to norovirus and worthy of a figure in this review? I think a better figure would be to describe what specific environmental materials that contain HBGAs and their ability to bind to norovirus.
Response: We have removed the Figure on HBGA-biosynthesis. Additional description was provided on the HBGA-norovirus interaction between line 84-93. A new table (Table 1) was included to summarize the presence of HBGA-like substances in various environmental matrices and their specific interactions with norovirus.
Comment: Figure 3: Having a figure dedicated to rotavirus seems outside of the scope of this review.
Response: In that study, we used rotavirus HAL1166 merely as a culturable surrogate for human norovirus because we are interested in the HBGA-virus specific interaction (line 155-162). By using rotavirus HAL1166, we had the additional advantage of distinguishing between specific and non-specific interactions because VP8* protein generation by trypsin treatment is needed for rotavirus HAL1166 to interact with HBGA. Therefore, we think rotavirus HAL1166 is a suitable surrogate for studying specific and non-specific interactions of norovirus with HBGA.
Minor:
Comment: There are many minor grammar errors throughout the text that need to be addressed.
Response: We have rechecked the manuscript for grammatical errors. We used the track changes function during the modification to highlight the changes.
Reviewer 2 Report
In the manuscript entitled ‘Specific interactions between human norovirus and environmental matrices: Effects on the virus ecology’ the authors provide an excellent literature review on the norovirus binding capabilities to HBGA-like substances and aim to highlight the usefulness and relevance of those interactions in environmental matrices. Overall, the content of the manuscript is highly relevant, the literature review is thorough and up to date and the content is appropriate to the journal. My comments below aim to enhance readability.
Major comments
1. However, the manuscript provide a thorough review on literature it does not provide any conclusions on the reviewed topics. I believe it would be beneficial to add a few sentences or a paragraph to the end of each section/subsection, which summarises the take-home message of the content of the section and highlights why the findings are relevant in an environmental setting.
2. Section 2: A figure showing the interactions between norovirus proteins and HBGA would be useful.
3. It would probably help readability if a summary table was added to the paper, mainly based on Section 3. It would show which cells/tissue has which HBGA-like substance and which norovirus strains bind to them. That would enable readers to have an overview by a glance, without going into details of each reviewed study.
4. Section 4.1: Such detailed summary on rotavirus interactions (lines 136-144, Figure 3) seems out of scope in a norovirus paper. I suggest to shorten/delete this part.
5. Section 4.3: This section contains three seemingly unrelated paragraphs and does not really address the title of the section. I suggest to rewrite this section and add more discussion on the effects of these interactions on viral ecology.
6. Lines 255-257: reference 90 should be discussed in the previous section.
Minor comments
There are minor grammar and style errors throughout the manuscript. Some of those are indicated in the enclosed file. I recommend a thorough proofreading prior to resubmission.

Author Response
Comment: However, the manuscript provides a thorough review on literature it does not provide any conclusions on the reviewed topics. I believe it would be beneficial to add a few sentences or a paragraph to the end of each section/subsection, which summarises the take-home message of the content of the section and highlights why the findings are relevant in an environmental setting.
Response: Thank you very much for your comments. We have modified the manuscript and added some new paragraphs to improve the readability of the manuscript (line 36-47, 255-261).
Comment: Section 2: A figure showing the interactions between norovirus proteins and HBGA would be useful.
Response: We have expanded our description about how HBGA-norovirus interaction occurs (line 84-93).
Comment: It would probably help readability if a summary table was added to the paper, mainly based on Section 3. It would show which cells/tissue has which HBGA-like substance and which norovirus strains bind to them. That would enable readers to have an overview by a glance, without going into details of each reviewed study.
Response: We included Table 1 in the modified manuscript as suggested.
Comment: Section 4.1: Such detailed summary on rotavirus interactions (lines 136-144, Figure 3) seems out of scope in a norovirus paper. I suggest to shorten/delete this part.
Response: We used human rotavirus HAL1166 merely as a surrogate for human norovirus because HAL1166 also specifically interacts with A-type HBGA. By using HAL1166, we also had the advantage of differentiating specific and non-specific interactions. The passage is shortened as suggested (line 155-162)
Comment: Section 4.3: This section contains three seemingly unrelated paragraphs and does not really address the title of the section. I suggest to rewrite this section and add more discussion on the effects of these interactions on viral ecology.
Response: We expanded and rearranged the complete 4.3 section (line 222-261)
Comment: Lines 255-257: reference 90 should be discussed in the previous section.
Response: We moved the reference in to section 4.3 (line 247-251)
Minor comments
Comment: There are minor grammar and style errors throughout the manuscript. Some of those are indicated in the enclosed file. I recommend a thorough proofreading prior to resubmission.
Response: We corrected the mistakes as suggested and rechecked the manuscript for grammatical errors. We used the track changes function during the modification to highlight the changes.